# Shared Genomic and Proteomic Contribution of Amyloid and Tau Protein Characteristic of Alzheimer’s Disease to Brain Ischemia

**DOI:** 10.3390/ijms21093186

**Published:** 2020-04-30

**Authors:** Ryszard Pluta, Marzena Ułamek-Kozioł, Sławomir Januszewski, Stanisław J. Czuczwar

**Affiliations:** 1Laboratory of Ischemic and Neurodegenerative Brain Research, Mossakowski Medical Research Centre, Polish Academy of Sciences, 02-106 Warsaw, Poland; mulamek@imdik.pan.pl (M.U.-K.); sjanuszewski@imdik.pan.pl (S.J.); 2Department of Pathophysiology, Medical University of Lublin, 20-090 Lublin, Poland; czuczwarsj@yahoo.com

**Keywords:** brain ischemia, Alzheimer’s disease, stroke, cardiac arrest, amyloid, tau protein, amyloid protein precursor, α-secretase, β-secretase, presenilins, gene expression, dementia

## Abstract

Post-ischemic brain damage is associated with the deposition of folding proteins such as the amyloid and tau protein in the intra- and extracellular spaces of brain tissue. In this review, we summarize the protein changes associated with Alzheimer’s disease and their gene expression (amyloid protein precursor and tau protein) after ischemia-reperfusion brain injury and their role in the post-ischemic injury. Recent advances in understanding the post-ischemic neuropathology have revealed dysregulation of *amyloid protein precursor, α-secretase, β-secretase, presenilin 1 and 2,* and *tau protein* genes after ischemic brain injury. However, reduced expression of the α-secretase in post-ischemic brain causes neurons to be less resistant to injury. In this review, we present the latest evidence that proteins associated with Alzheimer’s disease and their genes play a key role in progressive brain damage due to ischemia and reperfusion, and that an ischemic episode is an essential and leading supplier of proteins and genes associated with Alzheimer’s disease in post-ischemic brain. Understanding the underlying processes of linking Alzheimer’s disease-related proteins and their genes in post-ischemic brain injury with the risk of developing Alzheimer’s disease will provide the most significant goals for therapeutic development to date.

## 1. Introduction

Most studies on the consequences of cerebral ischemia have been conducted in rodents. Preferring rodent ischemia and reperfusion brain research is supported by high homogeneity due to inbred, low cost, availability, and similarity of the brain vascular system in rodents and humans. For several reasons, the hippocampus is the preferred brain sector for studying post-ischemic repercussions. First, the CA1 area of the hippocampus is a brain region very sensitive to ischemic episodes. Secondly, all regions of the hippocampus are involved in memory and spatial learning. Third, the hippocampus is a structure that shows identical changes in the brain after ischemia and Alzheimer’s disease.

Transient cerebral ischemia-reperfusion causes mass death of pyramidal neurons in the hippocampal CA1 region and in the third, fifth, and sixth layers of the cerebral cortex. In the above structures, necrotic and apoptotic neurons were mixed with damaged neurons within seven days after ischemic brain injury [1,2,3,4]. Within 6 months of recirculation, the number of damaged neurons decreased and the number of dead neurons increased. During survival longer than six months after cerebral ischemia, acute and chronic neuronal changes in ischemic resistant areas were observed in addition to acute neuronal death in ischemic sensitive areas. Changes occurred in areas of the brain that were not affected by primary ischemic changes, such as the CA2, CA3, and CA4 regions of the hippocampus [1,2,4]. Neuronal death along with a decrease in the level of acetylcholine in the hippocampus was noted after ischemia, suggesting that neuronal death was caused by a deficiency of neuronal excitation [5,6,7]. In addition, ultrastructural changes after ischemia were observed in hippocampus synapses [7,8,9]. Other studies have shown that an episode of cerebral ischemia leads to the induction of synaptic autophagy, which may be associated with loss of neurons in the hippocampus after transient cerebral ischemia [5,6,7,8,10,11,12]. Intracellular calcium increase post-ischemia [5] stimulates the activity of calpain in neurons whose target proteins are found in GABAergic and glutaminergic synapses [7]. Following brain ischemia, calpain cleaves pre- and postsynaptic proteins, and calpain-cleaved proteins ultimately contribute to the death of ischemic neuronal cells [13].

Changes in white matter and activation of neuroglial cells in the brain were observed in both humans and animals after ischemia [1,2,14,15,16,17,18,19,20,21,22,23]. In experimental models of transient brain ischemia, ischemia causes serious alterations in both the corpus callosum and subcortical white matter [2,17,18,24]. These alterations are consistent with the activation of neuroglial cells in the corpus callosum in the post-ischemic brain [25]. Late atrophy of the white matter of the brain manifested itself as advanced spongiosis. Ischemic changes in the brain showed signs of progressive neurodegeneration that developed slowly over a long period of time within recirculation after an episode of cerebral ischemia [2]. Brain autopsy carried out within 1–2 years after ischemia showed the features of hydrocephalus [1,2,26] with the widening of the ventricles and the subarachnoid space around the cerebral hemispheres [1]. During this time, general atrophy of the hippocampus with very narrow cerebral cortex was observed [1,2,4,26,27]. The final consequence of these changes is the development of dementia in experimental and clinical studies after cerebral ischemia [19,28,29,30,31,32].

In addition, it was noted that post-ischemic neurodegeneration processes occur not only in the acute phase of ischemia but last throughout the recirculation period [2]. The brain neurodegeneration profile that is observed after ischemia has common features with neurodegeneration in Alzheimer’s disease [4,33,34,35,36,37,38,39,40]. This confirms an increase in the blood-brain barrier permeability after ischemia for inflammatory cells and leaks of amyloid and tau protein from the blood to the brain tissue, which in turn probably leads to irreversible and progressive damage to the entire brain [17,18,20,32,41,42,43,44,45,46,47,48,49,50,51,52,53,54,55,56]. Understanding the deterioration of the mental state associated with brain neurodegeneration after ischemia sparked serious scientific debate. Therefore, the role of amyloid and tau protein as additional causative agents in the development of dementia after ischemia has recently been noticed [32,56,57]. Brain neurodegeneration due to ischemia-reperfusion has been found to be associated with the production and accumulation of folding proteins such as amyloid and tau protein [3,36,38,39,58,59,60,61]. We present here changes in proteins associated with Alzheimer’s disease and the expression of their genes (*amyloid protein precursor,* and *tau protein*) after ischemic-reperfusion injury of the brain and their role in post-ischemic neurodegeneration. New advances in understanding the possible development of post-ischemic neurodegeneration have revealed dysregulation of the *amyloid protein precursor*, *α-secretase, β-secretase, γ-secretase* and *tau protein* genes. In this review, we also present the latest evidence that Alzheimer’s disease-associated proteins and their genes play an important role in the progression of brain neurodegeneration after cerebral ischemia.

## 2. Amyloid in Post-Ischemic Brain

### 2.1. Dysregulation of Amyloid Associated Genes

In the CA1 area of the hippocampus, the expression of the *amyloid protein precursor* gene was below the control value 2 days post-ischemia (Table 1) [62]. Seven and thirty days following the episode of ischemia and reperfusion, the expression of the *amyloid protein precursor* gene was above the control value (Table 1) [62]. The expression of the *β-secretase* gene increased above the control value 2–7 days after ischemia in the CA1 area (Table 1) [62]. Thirty days post-ischemia, *β-secretase* gene expression was below control value (Table 1) [62]. In the CA1 area, the expression of *presenilin 1* and *2* genes increased during 2–7 days after ischemia (Table 1) [62]. In contrast, thirty days post-ischemia, the expression of *presenilin 1* and *2* genes was below the control value (Table 1) [62].

The statistical significance of changes in gene expression of the *amyloid protein precursor, β-secretase,* and *presenilin 2* was between 2 and 30, 2 and 7 and between 7 and 30 days after ischemia [62]. The statistical significance of changes in *presenilin 1* gene expression was between 2 and 30 and between 7 and 30 days after ischemia [62].

In the CA3 region 2, 7, and 30 days post-ischemia, the expression of the *amyloid protein precursor* gene was above control values (Table 2) [63]. In this area of the hippocampus, *α-secretase* gene expression was below control within 2, 7, and 30 days post-ischemia (Table 2) [63]. The expression of the *β-secretase* gene was below the control value post-ischemia in the hippocampal CA3 region for 2–7 days (Table 2). In contrast, 30 days post-ischemia, *β-secretase* gene expression was above control (Table 2) [63]. In the CA3 region, expression of the *presenilin 1* gene increased for 2–7 days post-ischemia (Table 2). Thirty days after cerebral ischemia, the expression of the *presenilin 1* gene was below the control value (Table 2) [63]. In this area, the expression of the *presenilin 2* gene was reduced for 2–7 days post-ischemia (Table 2). But thirty days after ischemia, the expression of the *presenilin 2* gene was above the control value (Table 2) [63].

The statistical significance of changes in expression of the *amyloid protein precursor* gene was between 2 and 7 and between 7 and 30 days post-ischemia [63]. No statistical significance was found during the entire period after ischemia in the *α-secretase* gene [63]. Statistically significant differences in the expression level of the *β-secretase* gene occurred between 2 and 30 days after ischemia [63]. The statistical significance of changes in gene expression of the *presenilin 1* and *presenilin 2* was between 2 to 30 and between 7 to 30 days after ischemia [63].

In the medial temporal cortex, the expression of the *amyloid protein precursor* gene was below the control value 2 days after ischemia (Table 3) [64]. In the above area, 7–30 days after ischemic injury, the expression of the *amyloid protein precursor* gene was above control values (Table 3) [64]. The *β-secretase* gene expression was above the control value within 2 days after ischemia (Table 3) [64]. Expression of the *β-secretase* gene was reduced in the medial temporal cortex 7–30 days post-ischemia (Table 3) [64]. The expression of the *presenilin 1* gene was lowered below the control value, while the *presenilin 2* gene was above the control value 2 days post-ischemia (Table 3) [65]. Seven days post-ischemia, the expression of the *presenilin 1* gene was reduced and the *presenilin 2* gene was increased (Table 3) [65]. Thirty days post-ischemia, the expression of the *presenilin 1* gene was above the control value and that of *presenilin 2* gene below the control value (Table 3) [65].

The statistical significance of changes in gene expression of the *amyloid protein precursor, β-secretase* and *presenilin 2* was between 2 and 7, and between 2 and 30 days after ischemia [64,65]. There was no statistically significant difference in expression levels of the *presenilin 1* gene throughout the whole observation time post-ischemia [65].

The results show that ischemic brain damage causes neuronal death in the hippocampus and medial temporal cortex in an amyloid-dependent mechanism, defining a new and very important process that ultimately regulates neuronal survival and/or death after ischemia (Table 1, Table 2 and Table 3) [62,63,64,65].

### 2.2. Dysregulation of Amyloid Associated mRNAs

Within 7 days after transient focal brain ischemia, the amyloid protein precursor mRNA increased by 150–200% [66,67]. In another study, only amyloid protein precursor mRNA containing the Kunitz-type protease inhibitor domain was observed in the post-ischemic period [68]. As a result of irreversible local brain ischemia, the mRNA of the amyloid protein precursor containing the Kunitz type protease inhibitor domain increased in cortex on day 21, but the total mRNA level did not change [69]. In addition, after reversible focal post-ischemic brain injury, the 751 and 770 amyloid protein precursor mRNA increased within 7 days [70]. Ovariectomized rats after local brain ischemia an hour after ischemia showed an increase in mRNA of the amyloid protein precursor [66]. The estrogen treatment used reduced the mRNA of the amyloid protein precursor in areas of ischemia [66].

In the non-amyloidogenic pathway, the amyloid protein precursor is metabolized by α-secretase. After experimental brain ischemia, mRNA α-secretase level and gene expression are reduced [63,71,72]. In the amyloidogenic metabolism, the amyloid protein precursor is cleaved by β- and γ-secretase to form β-amyloid peptide [34]. There is evidence that ischemia activates the expression, production, and activity of β-secretase [62,63,73,74,75,76]. Another study showed post-ischemic changes in the cortex and hippocampus at the mRNA level of three enzymes that metabolize the amyloid protein precursor: β-secretase, glutaminyl cyclase, and cathepsin B, whose levels increased rapidly [77]. Presenilin mRNA, which is induced by brain ischemia [78,79], is involved in the generation of β-amyloid peptide by the γ-secretase complex. An increase in presenilin 1 mRNA was observed in the hippocampal CA3 region and dentate gyrus in animal studies of post-ischemic brain injury [78]. Presenilin 1 mRNA had the highest level of expression on day 3 post-ischemia [78]. In another study, elevated presenilins mRNA levels after ischemia were found in the hippocampus, brain cortex, and striatum [79]. The maximum increase in presenilins mRNA was noted in the hippocampus and cortex. An increase in presenilin 1 and 2 mRNA was observed in the cortex within 1–8 days after ischemia [79]. In the hippocampus, presenilin 1 and 2 mRNA was upregulated in 4–8 days post-ischemia [79]. The above observations help to understand the progressive neuronal death after an episode of cerebral ischemia with reperfusion, massive accumulation of β-amyloid peptide, as well as the slow development of dementia with the phenotype of Alzheimer’s disease [2,28,29,30,31,52].

### 2.3. Changes in Amyloid Staining in Animal and Human Brain

In animals after brain damage due to ischemia-reperfusion, with survival up to 1 year, staining of β-amyloid peptide was revealed in the intra- and extracellular space of brain tissue [1,2,26,43,59,80,81,82,83,84,85,86,87,88,89,90,91,92,93,94]. Amyloid staining was observed after ischemia in neurons and neuroglial cells [1,85,89,95,96,97,98,99]. Observed astrocytes with massive amyloid accumulation in the cytoplasm may be involved in the development of glial scars [1,89,97,98,99]. In addition, reactive astrocytes with accumulated amyloid in the cytoplasm are probably involved in the pathological repair of post-ischemic brain tissue, accompanied by death of astrocytes [1,43,89,100,101]. After ischemia, amyloid staining was found in the periventricular and subcortical white matter [2,17,18]. It was found that the more intense the damage to white matter after ischemia, the more intense staining of amyloid in this area was [14,102]. The abovementioned changes were associated with the appearance of leukoaraiosis after ischemia in the brain [18]. Usually extracellular amyloid deposits occurred as very small dots or as diffuse amyloid plaques [1,2,43,47,48,51,52,87,89,103,104,105]. Deposition of β-amyloid peptide in the form of diffuse plaques in response to experimental ischemic brain injury is not a transient phenomenon, since it has been observed that diffuse amyloid plaques transform into senile plaques about 1-year post-ischemia [106]. Multifocal amyloid plaques have been reported in ischemic cortex, hippocampus, entorhinal cortex, corpus callosum and thalamus, and around the lateral ventricles. The accumulation of β-amyloid peptide in ischemic neurons and astrocytes indicates the pathological role of amyloid in post-ischemic neurodegenerative processes of the brain [43,83,98,99,103,104]. These data indicate that the increased accumulation of β-amyloid peptide in the brain after ischemia may be responsible for secondary neurodegenerative processes that worsen post-ischemic outcome through progressive neuronal loss [2,26,29,45,57,87,93,94,107,108,109]. It is noted that after ischemia, amyloid is formed as a result of damage and death of neurons [82] and its neurotoxic activity promotes the slow development of brain atrophy and dementia of the Alzheimer’s disease phenotype [110]. Amyloid is a neurotoxic molecule and post-ischemia initiates pathological processes in neurons, astrocytes, microglia and oligodendrocytes that affect neurons and neuroglial cells, causing them to die [94,111].

Accumulation of amyloid in various brain structures was noted during autopsy of human ischemic brains [112,113,114,115]. After ischemia, diffuse and senile amyloid plaques have been shown in the arterial border zones and areas sensitive to ischemia [112,114]. In addition, it was noted that amyloid was most often present in the middle layers of the brain cortex, which are very susceptible to ischemia. Another study found mass accumulation of amyloid in neurons and perivascular areas in the brain post-ischemia due to cardiac arrest with survival of 1 month [113]. In this study, senile amyloid plaques were described in two cases. The hippocampal and cortical neurons, as well as epithelial and ependymal cells, were intensely stained for amyloid. Gray and white matter cerebral vessels were surrounded by amyloid deposits that were mainly cuff-shaped [113]. In some brains, the walls of cortical and meningeal vessels were intensely stained with amyloid. According to another study, β-amyloid peptide 1–40 and 1–42 was found in the human hippocampus post-ischemia [115]. Intensive amyloid staining suggests its involvement in the progression of neurodegeneration after ischemia and the development of dementia with the phenotype of Alzheimer’s disease. The results show that ischemic brain damage causes amyloid-dependent hippocampal neuronal death, thus defining a new, very important mechanism that ultimately determines survival and/or death of neurons after ischemia (Figure 1).

### 2.4. Blood-Brain Barrier and Amyloid in the Blood

In patients during 4 days after brain ischemia due to cardiac arrest, the increase in blood β-amyloid peptide 1–42 was approximately 70-fold compared with control [54]. The value of growth correlated negatively with the clinical outcome post-ischemia [50,54,56]. These studies provide direct evidence that human brain ischemia causes an increase in the blood level of β-amyloid 1–42 peptide. The data indicate that acute cerebral ischemia may trigger an amyloidogenic process in Alzheimer’s disease. The level of serum amyloid growth probably reflects the degree of brain damage following an ischemic episode [50,54,56]. In addition, the relationship between elevated blood amyloid level and clinical outcomes suggests a direct relationship between an ischemic episode and a level of β-amyloid peptide 1–42, which is not secondary in the patients studied [54].

In addition, a receptor for advanced glycation end products was found in the brains of patients after ischemia due to cardiac arrest in the epithelial cells of the choroid plexus and in the lining ependymal cells adjacent to the brain ventricles [116]. The above cells form the blood-cerebrospinal fluid barrier and the cerebrospinal-brain barrier. Staining for amyloid was observed in the walls of the choroid plexus blood vessels and in the basal membrane of the choroid plexus epithelium [116]. Amyloid has been reported in cytoplasmic vacuoles of many epithelial and ependymal cells of the choroid plexus. Data has shown that choroid plexus epithelium and ependymal cells equipped with a receptor for advanced glycation end products can play a significant role in the transport and accumulation of amyloid in brain tissue. In addition, amyloid accumulation around the blood-brain barrier vessels suggests that β-amyloid peptide is derived from blood. Evidence supporting this hypothesis comes from clinical studies that showed an increase in blood amyloid level in patients following ischemic brain injury [50,54,56]. In addition, experimental studies point to the passage of human amyloid from the blood through the ischemic blood-brain barrier [45,46,47,86]. The receptor for advanced glycation end products may be the main therapeutic target in post-ischemic brain amyloidosis.

## 3. Tau Protein in Post-Ischemic Brain

### 3.1. Dysregulation of the Tau Protein Gene

A relationship has been demonstrated between hippocampal CA1 neuronal damage and *tau protein* gene expression after 10 min of global cerebral ischemia in rats, with survival 2, 7, and 30 days post-ischemia [117]. In CA1 neurons, *tau protein* gene expression increased above the control value on the second day after cerebral ischemia (Table 1) [117]. On the seventh and thirtieth day of recirculation after an ischemic episode, gene expression was below the control values (Table 1) [117]. The statistical significance of changes in *tau protein* gene expression in rats was between 2 and 7 and 2 and 30 days after ischemia [117].

In the CA3 region of the hippocampus, the expression of the *tau protein* gene after ischemia with a survival time of 2 days was below control values (Table 2) [63]. But 7–30 days after ischemia, *tau protein* gene expression was higher than control values (Table 2) [63]. The changes were statistically significant between days 2 and 7 and between days 2 and 30 after ischemia [63].

The results show that ischemic brain damage causes neuronal death in the hippocampus in a tau protein-dependent mechanism, defining a new and very important process that ultimately regulates neuronal survival and/or death after ischemia (Figure 2) [63].

### 3.2. Changes in Tau Protein Staining in Animal and Human Brain

A common appearance of immunoreactive tau protein neurons and neuroglial cells was found in human and experimental post-ischemic hippocampus, thalamus, and cortex [58,90,118,119,120,121,122,123,124,125]. Some neurons were also labeled with tau protein antibodies after cerebral ischemia in humans due to cardiac arrest with 1 month survival [113]. After focal cerebral ischemia, tau protein staining was also noted in microglia [125]. The evidence presented indicates that some neuronal cells show changes in tau protein during post-ischemic brain injury [121] that may be associated with the degree of development of ischemic neuron death (Figure 2) [124].

### 3.3. Blood-Brain Barrier and Tau Protein in the Blood and Cerebrospinal Fluid

Increased level of amyloid and tau protein after brain ischemia in serum [50,53,54,55,56,126,127,128,129,130,131] and cerebrospinal fluid [131,132] combine the pathology of amyloid and tau protein with ischemic blood-brain barrier failure [133]. In addition, oxidative stress [134] and neuroinflammation [20,22,23] induced by increased permeability of the blood-brain barrier can initiate phosphorylation of tau protein and development of neurofibrillary tangles after ischemia [3,75,135,136,137,138,139]. Increased plasma tau protein [53,55] may cross the ischemic blood-brain barrier, and blood-derived tau protein may increase brain pathology after ischemia [140]. Ischemic brain injury with insufficient blood-brain barrier [17,18,42,43,45,48,51] initiates tau protein phosphorylation [75,139,141,142,143] and phosphorylated tau protein can cause damage to blood-brain barrier, leading to harmful feedback [133]. The above suggests that brain damage as a result of an ischemic episode with reperfusion may play an important role in increasing plasma tau protein level [53,55,140].

Increased level of tau protein in human blood was noted after ischemia due to cardiac arrest with two peaks on days 2 and 4, indicating the progression of neuronal changes [53,55]. The observed two-stage kinetics of the increase in the level of soluble tau protein in plasma is consistent with two types of neuronal death—firstly by necrosis and secondly by apoptosis [55]. It seems very likely that the profiles reflect the time course of acute and delayed ischemic damage or death of neurons [55]. The above studies suggest that the level of tau protein in human blood can be used as a prognostic element of the neurological outcome after ischemia [53,55].

### 3.4. Tau Protein Hyperphosphorylation

After transient local and complete cerebral ischemia, tau protein dephosphorylation was noted [120,121,144,145]. But in another study after transient global cerebral ischemia due to cardiac arrest, the tau protein was gradually re-phosphorylated [145]. In addition, a site-specific hyperphosphorylation of tau protein was observed in animals after transient focal cerebral ischemia [75]. At the time of neuronal death in the hippocampal CA1 region after forebrain ischemia in gerbils, serine 199/202 hyperphosphorylation of tau protein was synchronized with GSK3, CDK5, and MAP kinases [146]. New data indicate that in the brain after ischemia with reperfusion, modifications of tau protein by hyperphosphorylation are comparable to those found in Alzheimer’s disease and are accompanied by apoptosis [137,138,141,147]. The above observations indicate that in ischemic brain injury, apoptosis is directly related to tau protein hyperphosphorylation. Another study showed the production of paired helical filaments of tau protein after ischemia in animals [148]. Additional studies provided data that the ischemia-reperfusion event of the brain was involved in the development of neurofibrillary tangle-like [136,137,138]. Neurofibrillary tangles were found after human cerebral infarction [135]. In addition, the combination of cerebral ischemia with hyperhomocysteinemia in animals resulted in neuronal changes of the cerebral cortex and hippocampus caused by tau protein hyperphosphorylation [143]. This study revealed an approximately 700-fold increase in the number of neurons with hyperphosphorylated tau protein in the brain after ischemia compared to control [143]. Dysfunctional tau protein increases post-ischemic brain damage through iron export [149] and self-excitotoxicity (Figure 2) [5,150].

## 4. Discussion

This review features the response of *amyloid* and *tau protein* genes and their products to post-ischemic brain injury (Figure 1, Figure 2 and Figure 3). Data showed that after ischemia, overexpression of the *amyloid protein precursor* gene began and correlated with the massive increase of soluble amyloid in blood (Figure 1 and Figure 3) [50,54,56] and intra- and extracellular space [2,43] as well as with development of diffuse and senile plaques [113]. The data also revealed that after ischemia, overexpression of the *tau protein* gene in the brain began and correlated with the massive increase of soluble tau protein in blood (Figure 2 and Figure 3) [53,55] and extracellular space [151], as well as with the hyperphosphorylation of tau protein [3,61]. Increased expression of the *amyloid* and *tau protein* genes was parallel to the onset of delayed neuronal death after ischemic brain injury (Figure 3) [1,2,26]. The increase in brain and serum amyloid levels [2,43,50,54,56] was associated with a similar increase in brain and blood levels of tau protein after ischemia [3,53,55,63], and these changes predict a worse clinical outcome. Ischemia-induced increase in *tau protein* gene expression was parallel to overexpression of *caspase 3* gene and caspase plays an important role in neuronal death (Figure 3) [11,12,152]. The data showed that activated caspase positively correlates with the development of neurofibrillary tangles [3]. In addition, cognitive deficits are negatively correlated with levels of amyloid and tau protein [3,39]. Data suggest that when tau protein is ischemically translated, its hyperphosphorylation increases, which means that hyperphosphorylation of tau protein is driven by the substrate, and transcription levels are identical to protein levels (Figure 3) [137,138]. Another study showed elevated Cdk5 levels in animals exposed to local reversible cerebral ischemia, confirming the above observations [137]. An increase in tau protein hyperphosphorylation may be a consequence of increased translation or inhibition of its degradation or blocked clearance. Data show that post-ischemic brain injury activates neuronal changes and death in the brain dependent on amyloid and tau protein, thus determining a new and important way to regulate neuron survival and/or death after ischemia (Table 1 and Table 2, Figure 1, Figure 2 and Figure 3). Induced pathological changes such as oxidative stress, apoptosis, autophagy and excitotoxicity, neuroinflammation by amyloid and tau protein determine their potential pathological mechanisms in the brain after ischemia (Figure 1, Figure 2 and Figure 3).

The presented facts confirm the opinion that brain damage after ischemia with reperfusion plays an important role in the pathological behavior of amyloid and tau protein in brain tissue and plasma (Figure 1, Figure 2 and Figure 3). Expression of amyloid and tau protein genes and their brain and blood protein levels that are increased after ischemia (Table 1 and Table 2) (Figure 1, Figure 2 and Figure 3) [3,50,53,54,55,56,62,63,64,117], are involved in the development of neuropathology characteristic of Alzheimer’s disease. One study provided evidence that the regional distribution of tau protein from neuropil to the neuronal body after cerebral ischemia was similar to that found in Alzheimer’s disease [141]. It is highly likely that the modified amyloid and tau protein additionally increase ischemic damage and/or neuronal death post-ischemia (Figure 1, Figure 2 and Figure 3). The above evidence allows us to identify acute and chronic processes during neuronal death and the development of slow and progressive brain atrophy after ischemia with dementia with the Alzheimer’s disease phenotype (Figure 1 and Figure 2) [1,2,19,28,29,30,31,32,153,154]. After cerebral ischemia in humans, the increase in plasma levels of amyloid and tau protein negatively correlated with clinical outcome, which reflected the degree of brain damage [50,53,54,55,56]. It seems that post-ischemic brain injury promotes the development of irreversible neurodegeneration of the Alzheimer’s disease type with massive neuronal loss [1,2], neuroinflammation [20,22,23], changes in white matter with general brain atrophy [1,2,26,27], and accumulation of amyloid [2,43,52] and dysfunctional tau protein [3,61,63,117]. Although significant progress has recently been made in studying the pathogenicity of amyloid and tau protein after ischemia, key mechanisms involved in irreversible ischemic brain neurodegeneration induced by amyloid and tau protein are still unknown. Post-ischemic brain damage has also been shown to induce neuronal death in association with amyloid and tau protein (Figure 1, Figure 2 and Figure 3) [39,62,117], defining a new and important way of regulating neuronal survival or death. The relationship between amyloid and tau protein associated with Alzheimer’s disease and experimental cerebral ischemia and ischemic stroke in humans appears to be significant.

According to the scientific observations, it can be stated that transient ischemic brain injury modifies tau protein and amyloid at both gene and protein levels (Figure 3), leading to development of amyloid plaques [43,106,113] and the accumulation of tau protein as neurofibrillary tangles in the brain tissue [39,117,135,137,148]. The conclusions presented from the exploration of Alzheimer’s disease-related tau protein and amyloid and their genes in ischemic brain injury, which are partly associated with neuronal death by the development of neurofibrillary tangles and amyloid plaques (Figure 1, Figure 2 and Figure 3), are key to improving treatment of irreversible ischemic neurodegeneration [155,156,157]. Since the decreasing importance of tau protein and amyloid in the etiology of Alzheimer’s disease is proposed [158,159,160] and it is believed that the deposition of tau protein and amyloid is not the reason of Alzheimer’s disease, as presented in the NIA-AA Research Framework: towards the true explanation of Alzheimer’s disease [161], in this situation we need more innovative investigation in this field. Therefore, the animal models of brain ischemia with reperfusion used in the exploration of Alzheimer’s disease seem to be a useful new methodology to clearing up the role of folding proteins and their genes in neurodegeneration of brain ischemia and Alzheimer’s disease.

Although the role of ischemia in amyloid changes and tau protein hyperphosphorylation is generally complex and requires further research, and amyloid and tau protein are a relatively underestimated pathological factors in the brain after ischemia in animals and humans, we have reason to believe that determining the role of these molecules in brain ischemia can help us understand the basis for developing a new treatment goals for ischemic stroke in a human clinic [155,156,157]. Everything indicates that the regulation of amyloid and tau protein activity can be considered as a potential new therapeutic target in ischemic stroke [155,156,157,162,163].

## 5. Conclusions

Data indicate genomic and proteomic changes of amyloid and tau protein in post-ischemic hippocampus and medial temporal cortex. Thus, two-sided damage to the above-mentioned regions causes a short-term memory deterioration, which leads to the inability to create new memories. It is well-known that amyloid and hyperphosphorylated tau protein are closely associated with neurodegeneration and cognitive impairment in Alzheimer’s disease. However, further research is needed to determine whether damage and death of neurons in the hippocampus and medial temporal cortex are causative events or independent consequences of ischemia occurring in parallel and leading to the development of neuropathology and dementia after ischemia of the nature of Alzheimer’s disease. It appears that the prevention of ischemic brain damage and early treatment of ischemic stroke may have important implications for the development of Alzheimer’s disease and deserve further research. Thus, animal models of cerebral ischemia appear to be a useful experimental approach for determining the role of genes and proteins directly or indirectly associated with Alzheimer’s disease. In-depth research into the shared genetic and protein mechanisms associated with these two neurological diseases can accelerate the current understanding of the neurobiology of cerebral ischemia and Alzheimer’s disease, as well as conduct future research on cerebral ischemia or Alzheimer’s disease in new directions.

## Figures and Tables

**Figure 1 ijms-21-03186-f001:**
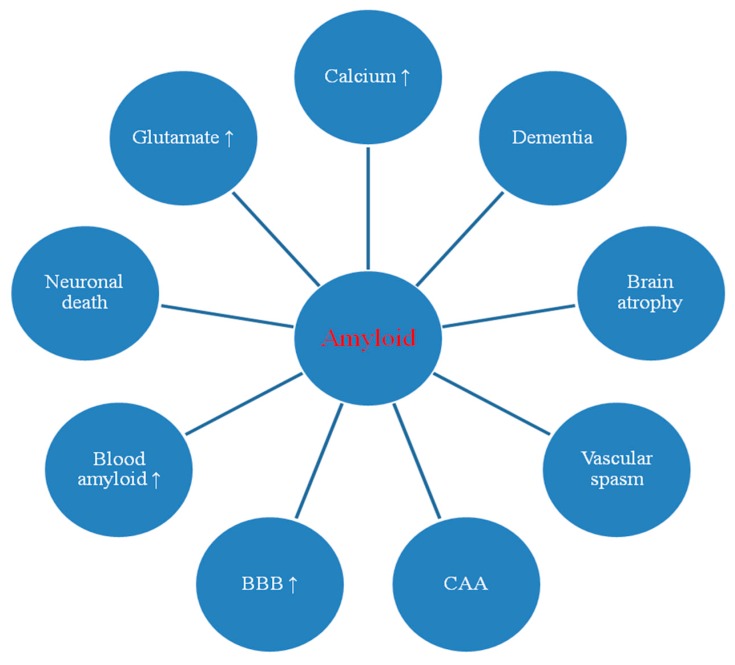
Potential pathological role of amyloid during ischemia-reperfusion brain injury. BBB: blood-brain barrier; CAA: cerebral amyloid angiopathy; ↑: increase.

**Figure 2 ijms-21-03186-f002:**
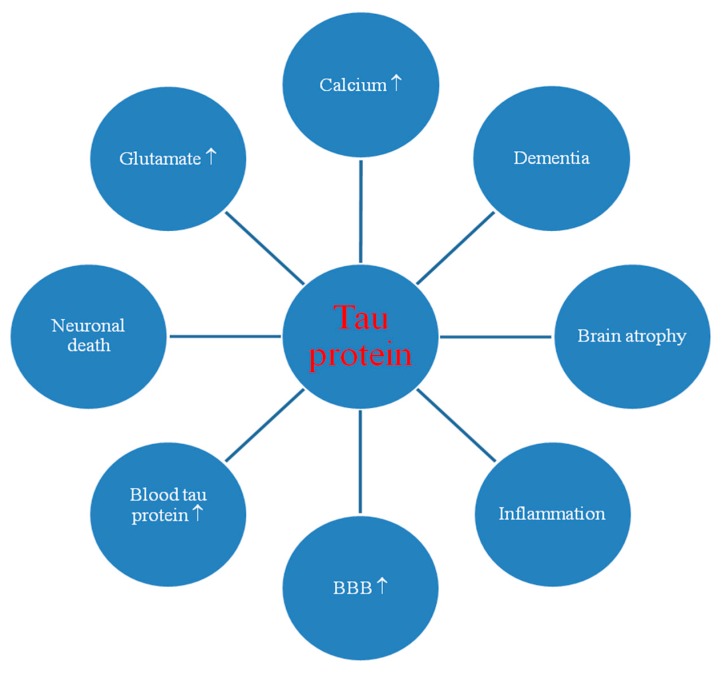
Potential pathological role of tau protein during ischemia-reperfusion brain injury. BBB: blood-brain barrier. ↑: increase.

**Figure 3 ijms-21-03186-f003:**
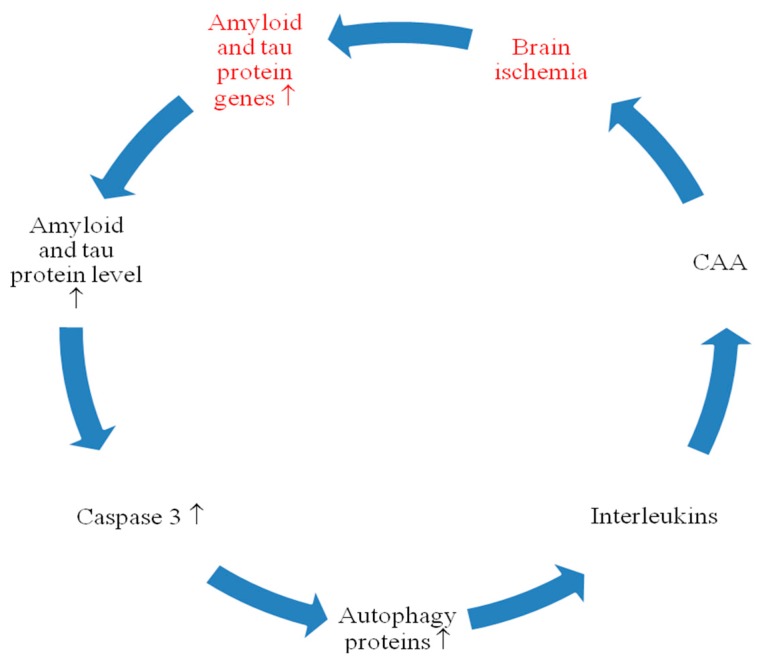
Cross talk between Alzheimer’s disease-associated proteins and their genes after brain ischemia. CAA: cerebral amyloid angiopathy. ↑: increase.

**Table 1 ijms-21-03186-t001:** Changes in the expression of Alzheimer’s disease-associated genes in the CA1 area of hippocampus at different times after experimental brain ischemia [62].

	Survival	2 Days	7 Days	30 Days
Genes				
*APP*		↓	↑	↑
*BACE1*		↑	↑	↓
*PSEN1*		↑	↑	↓
*PSEN2*		↑	↑	↓
*MAPT*		↑	↓	↓

Expression: ↑ increase; ↓ decrease. Genes: *APP-amyloid protein precursor*, *BACE1-β-secretase*, *PSEN1-presenilin 1*, *PSEN2-presenilin 2*, *MAPT-Tau protein*.

**Table 2 ijms-21-03186-t002:** Changes in the expression of Alzheimer’s disease-associated genes in the CA3 area of hippocampus at different times after experimental brain ischemia [63].

	Survival	2 Days	7 Days	30 Days
Genes				
*APP*		↑	↑	↑
*ADAM10*		↓	↓	↓
*BACE1*		↓	↓	↑
*PSEN1*		↑	↑	↓
*PSEN2*		↓	↓	↑
*MAPT*		↓	↑	↑

Expression: ↑ increase; ↓ decrease. Genes: *APP-amyloid protein precursor*, *ADAM10–α-secretase*, *BACE 1-β-secretase*, *PSEN1-presenilin 1*, *PSEN2-presenilin 2*, *MAPT-Tau protein*.

**Table 3 ijms-21-03186-t003:** Changes in the expression of Alzheimer’s disease-associated genes in the medial temporal cortex at different times after experimental brain ischemia [64,65].

	Survival	2 Days	7 Days	30 Days
Genes				
*APP*		↓	↑	↑
*BACE1*		↑	↓	↓
*PSEN1*		↓	↓	↑
*PSEN2*		↑	↑	↓

Expression: ↑ increase; ↓ decrease. Genes: *APP-amyloid protein precursor*, *BACE1-β-secretase*, *PSEN1-presenilin 1*, *PSEN2-presenilin 2*.

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
