# Peer review of "Shared Genomic and Proteomic Contribution of Amyloid and Tau Protein Characteristic of Alzheimer’s Disease to Brain Ischemia"

_ijms, 2020, doi:10.3390/ijms21093186_

Round 1

Reviewer 1 Report

A review “Shared Genomic and Proteomic Contribution of Amyloid and Tau Protein  Characteristic of Alzheimer’s Disease to Brain Ischemia” by Pulta et al.  summarizing the contribution of amyloid proteins originally associated with neurodegenerative diseases in the post-ischemic injury.  Overall the information provided in this review is important as well as valuable to general readers. The discussion and conclusion section is well written however the manuscript has some shortcomings in terms of presentation/interpretation of data and that need to be addressed. More specific comments are mentioned below.

  1. Restructure the tables to accommodate them together. It will be great to have the reference there as well.
  2. Line 99-105; this paragraph needs to modify to make it more clear. Since APP and b-secretase both show changes at the same time interval, they can be club together irrespective of their up- or down-regulation. Similarly, it can be used for other genes as well wherever duration is the same.
  3. Authors have discussed the changes in gene expression (of amyloid or tau protein) patterns and their significant differences in different sites in brains post-ischemia in sections 2 and 3. They should mention the relevance or significance of those changes at least briefly or indicate that it is discussed separately in the discussion section.
  4. I will advise authors to mention the reason for selectively using the genes (like BACE1-β-secretase, was used for CA1 but not ADAM10–α-secretase and so on).
  5. Line 190- check space between words.
  6. It is not clear to me whether the increased level of amyloid induces the various metabolic consequences as mentioned in figure 2 or decrease level. For example, the level of APP gene was down on day 2 and increased on 7/30 days time period in CA1 whereas in CA3 it was increased after 2 days; do these pathological changes reflect equally in all brain parts? Similarly, Figure 2, Do these changes are the result of an increased or decreased level of Tau protein which is again differentially changed in various brain parts.
  7. Correct reference number 61.

Reviewer 2 Report

This is an exceptional review in which the authors use complementary Genomic and Proteomic Contribution approaches to understand Amyloid and Tau Protein pathogenic Characteristics of Alzheimer’s Disease to Brain Ischemia. The review design is straightforward; the literature  are strong and support the conclusions. I firmly believe that the findings reported here will have a major impact on the field and as such I fully support the publication of these data.

To improve the manuscript further, the authors could also consider the similar studies like PMID: 30893872, PMID: 30453058; PMID: 31595293 tau and Amyloid together will be more added value for this review.
